# Field Campaign Evaluation of Sensors Lufft GMX500 and MaxiMet WS100 in Peruvian Central Andes

**DOI:** 10.3390/s22093219

**Published:** 2022-04-22

**Authors:** Jairo M. Valdivia, David A. Guizado, José L. Flores-Rojas, Delia P. Gamarra, Yamina F. Silva-Vidal, Edith R. Huamán

**Affiliations:** 1Instituto Geofísico del Perú, Lima 15012, Peru; jvaldivia@igp.gob.pe (J.M.V.); alejandroguivi@gmail.com (D.A.G.); 2Universidad Nacional del Centro del Perú, Huancayo 12006, Peru; d.gamarra@uncp.edu.pe (D.P.G.); ehuaman@uncp.edu.pe (E.R.H.); 3Instituto Nacional de Investigación en Glaciares y Ecosistemas de Montaña, Lima 15038, Peru; fsilva@inaigem.gob.pe

**Keywords:** meteorological instruments, drop size distribution, DSD, Huancayo Observatory, Peruvian Central Andes

## Abstract

The research presents the inter-comparison of atmospheric variables measured by 9 automatic weather stations. This set of data was compared with the measurements of other weather stations in order to standardize the values that must be adjusted when taken to different areas. The data of a set of a total of 9 GMX500, which measures conventional meteorological variables, and 10 WS100 sensors, which measures precipitation parameters. The automatic stations were set up at the Huancayo Observatory (Geophysical Institute of Peru) for a period of 5 months. The data set of GMX500 were evaluated comparing with the average of the 9 sensors and the WS100 was compared with an optical disdrometer Parsivel${}^{2}$. The temperature, pressure, relative humidity, wind speed, rainfall rate, and drop size distribution were evaluated. A pair of GMX500 sensors presented high data dispersion; it was found found that the errors came from a bad configuration; once this problem was solved, good agreement was archived, with low RMSE and high correlation. It was found that the WS100 sensors overestimate the precipitation with a percentage bias close to 100% and the differences increase with the greater intensity of rain. The drop size distribution retrieved by WS100 have unrealistic behavior with higher concentrations in diameters of 1 mm and 5 mm, in addition to a flattened curve.

## 1. Introduction

In this work, the WS100 (Lufft, OTT HydroMet Fellbach GmbH, Fellbach, Germany) and GMX500 (MaxiMet, Gill Instruments Limited, Lymington, UK) are used as part of a weather station, which would have the objective of covering most of the measurements necessary in research related to meteorology. Through a compact, automated and energy-efficient design, the WS100 and GMX500 are presented as one of the best options to carry out studies in areas of complex topography where data are scarce, such as the Andes. WS100 is a rain sensor, which uses a 24 GHz Doppler radar to measures all kind of precipitation with the possibility of retrieve the Drop Size Distribution (DSD). GMX500 is an advanced compact weather station which uses proved technology to measure wind speed and direction, humidity, temperature, and pressure, with no moving parts. The MaxiMet has a broad range of compact weather stations designed and manufactured by Gill Instruments (http://gillinstruments.com, accessed on 4 February 2022), several of these instruments has been used previous research (e.g., [1,2,3,4]) showing that they are reliable instruments. On the other hand, the Lufft is a brand of OTT Hydromet (www.ott.com, accessed on 4 February 2022) and has been developing and producing professional components for climate and environment for more than 135 years. Some OTT instruments such as the optical disdrometer Parsivel${}^{2}$ have been extensively used and evaluated (e.g., [5,6,7,8,9,10]). However, the WS100 rain sensor practically unknown in the scientific world and there are no studies evaluating how reliable its DSD retrievals are, as it is the case with the Parsivel${}^{2}$.

There is interest in being able to use the WS100 instead of the Parsivel${}^{2}$ since the WS100 sensor is much cheaper than the Parsivel${}^{2}$ and, based on information provided by the manufacturers, it is capable of providing similar information [11]. The Particle Size Velocity (Parsivel) is an optical disdrometer that measures both, size and velocity of the hidrometeors [12]. OTT then presented a second version of the Parsivel called Parsivel${}^{2}$, providing some improvements over its predecessor. Tokay et al. [5] evaluated the Parsivel${}^{2}$ with other types of disdrometers finding good agreement especially in middle drop diameters (0.5 to 4 mm). Compared to rain gauge the Parsivel${}^{2}$ had a 6% absolute bias in the event rain totals. Further comparisons of the Parsivel${}^{2}$ with 2D-video disdrometers have shown that the Pasivel${}^{2}$ underestimates drops smaller than 0.76 mm and overestimates drops greater than 4 mm [13], and corrections have been proposed to improve data quality [6,7]. Valdivia et al. [9] compared the Parsivel${}^{2}$ with tipping bucket rain gauges and a couple of radar profilers in the central Andes (3315 m MSL), founding a systematic underestimation in the event rain totals (18% bias and 19% absolute bias). However the Parsivel${}^{2}$ measurements are even more reliable than those provided by radar, and DSD retrievals are necessary for the radar algorithms optimization [14].

In order for the WS100 and GMX500 to be used with complete confidence in the future research studies, an analysis of the quality of his data is necessary. In the present study, an inter-comparison is made between 10 compact stations, which are assembled with the WS100 rain sensor and GMX500 weather station. The experimental study also includes a Parsivel${}^{2}$ which is used as reference for rain measurements and DSD evaluation. The technical details of the instruments are provided in Section 2. The results of the inter-comparison between instruments are shown in Section 3. The discussions is presented in Section 4, followed by the conclusions in Section 5. This work aims to show what is the real performance of these instruments as well as highlight their strengths and weaknesses.

## 2. Materials and Methods

### 2.1. Site of Measurements

The measurements of the meteorological variables in this study were carried out at the Huancayo Observatory (12.03° S, 75.32° W, 3315 m MSL), which is located in the Andes between Cordillera Occidental and Huaytapallana Cordillera, Peru. The data was collected from 18 October 2020 to 31 January 2021. 10 automatic stations were installed at a horizontal distance of 3.5 m and the instruments at 2 m from the ground surface in a north-south direction on tripods placed in the study area (see Figure 1). Of all the stations, 9 were acquired by the “CEPREANDES” project (sensors 1–9) and 1 station (sensor 10) was added in October 2020 as part of a collaboration of the “CEPREANDES” project.

The study was carried out in a total area of 49 m${}^{2}$. The precipitation measurements come from a rain sensor WS100 and Pasivel${}^{2}$, the temperature, pressure, relative humidity and wind velocity from the GMX500. The assembly of each station was carried out meticulously according to the guides and manual for the installation of automatic weather stations.

### 2.2. Instrumentation

#### 2.2.1. Compact Automatic Stations

The 10 compact automatic stations are assembled in the same way. Cables for the installation of sensors, battery, solar panel and datalogger. The cables connect the rain sensor to the datalogger, like the meteorological sensor, a cable from the solar panel goes directly to the datalogger and another to the battery. Figure 2 shows the components of the stations. The detailed description of each component in the automatic station is presented as follows:Weather Sensor GMX500 MAXIMET GILL series (Figure 2a): MaxiMet is an compact weather station (Gill Instruments Ltd., Hampshire, UK). Measurement parameters: wind, temperature, humidity, pressure, compass, GPS (optional).WS100 Lufft Precipitation Sensor (Figure 2b): A Smart Disdrometer (G. Lufft Mess- und Regeltechnik GmbH, Fellbach, Germany), using a 24 GHz Doppler radar, it measures the speed of all forms of condensed water. These include rain, freezing rain, hail, snow, and sleet.Campbell Scientific Box (Figure 2c): The ENC16/18 cabinet (Campbell Scientific, Inc., Logan, UT, USA), the back plate of the ENC16/18 is pre-drilled with half-inch holes in the center.Stainless steel tripod (Figure 2d): The CM110 is a 10-foot instrumentation tripod (Campbell Scientific, Inc., Logan, UT, USA) that supports mounting of sensors, mounts, solar panels, and environmental enclosures.Datalogger CR310—CAMPBELL SCIENTIFIC (Figure 2e): The CR310 is a low-cost, compact, multipurpose measurement and control datalogger (Campbell Scientific, Inc., Logan, UT, USA) that has an integrated 10/100 Ethernet port and removable terminal blocks.Automatic station battery Ritar 12 v 65 Ah AGM (Figure 2f): The Ritar brand solar gel battery (Ritar International Group, Shenzhen, China) has a capacity of 65 Ah and has been designed for use in small and medium-power solar installations.SP30 solar panel (Figure 2g): The SP30 is a 30 W solar panel (Campbell Scientific, Inc., Logan, UT, USA). It is commonly used in systems that have higher than average power requirements, or in high mountain stations. It connects to our power supplies or regulators to charge the battery, and thus allows unattended operation of our systems in remote locations without the need for 220 Vac.

#### 2.2.2. The OTT Parsivel${}^{2}$

The OTT Parsivel${}^{2}$ is alaser optical disdrometer. The unit works on the principle of extinction and measures the particles precipitated from the shadow they generate when they pass through a laser sheet [12]. From the raw data obtained, the amount and intensity of rainfall, visibility conditions, kinetic energy and radar reflectivity of rainfall are calculated. The raw output provides the drops numbers in a 32 × 32 size versus velocity array. The size range is 0 to 25 mm, and the class width increase from 0.125 to 3 mm. Due to the low signal to noise ratio the first two size classes are left empty. The minimum detectable size is 0.25 mm. The fall velocity range is from 0 to 20 m s${}^{-1}$, and the class width increases with the velocity.

### 2.3. Data Analysis

Data from the GMX500 and WS100 of the compact automatic station are analyzed separately. The GMX500 data are inter-compared using the average of all stations as a reference, excluding the sensor under evaluation. The GMX500’s variables to be evaluated are: temperature, pressure, relative humidity, and wind intensity. The coefficient of determination (R${}^{2}$), the root mean square error (RMSE) and a linear regression are calculated for each station. WS100 data is evaluated using Parsivel${}^{2}$ as a reference. The total accumulated rainfall per event, the rainy minutes, the rain intensity, and the drop size distribution (DSD) measured for each event are evaluated separately. To be consistent with the metrics from previous studies (i.e., [5,9]), the statistics used in the rainfall evaluation are: the bias, relative bias, absolute relative bias and Pearson correlation.

## 3. Results

### 3.1. Inter-Comparison of GMX500

#### 3.1.1. Temperature

The first inter-comparison made between GMX500 sensors is temperature. The average of the 9 sensors was used as a reference to evaluate each sensor. Figure 3 shows the inter-comparisons of each sensor, which are enumerated in each panel. It can be seen that sensors 1 and 2 differ considerably from the other sensors (Figure 3a,b). While almost all sensors have an R${}^{2}$ greater than 0.98, sensors 1 and 2 have an R${}^{2}$ of 0.55 and 0.96, respectively. Sensor 1 has a higher dispersion of its data, its RMSE is 2.98 °C (Figure 3a), followed by Sensor 2 with 0.87 °C (Figure 3b), then Sensor 9 with an RMSE of 0.6 °C (Figure 3i). The rest of the sensors (i.e., 3–8) have an RMSE between 0.45 and 0.53 °C (Figure 3c–h). The evaluation of the linear regression indicates that all the sensors have a tendency close to the ideal (i.e., intercept equal to 0 and slope equal to 1), except, sensor 1 whose intercept is 3.08 and the slope is 0.761 (Figure 3a). The other sensors had an intercept less than 0.7 and the slope greater than 0.95 (Figure 3b–i). The statistical results of all the sensors are shown in Table 1.

#### 3.1.2. Pressure

Pressure inter-comparison shows great similarity between all instruments to the naked eye. The inter-comparison of all instruments is shown in Figure 4. The R${}^{2}$ are very close to 1 in all cases. Sensors 1 and 2 have a slightly lower R${}^{2}$ than the rest (0.97 and 0.97, respectively), while all other sensors exceed 0.99. It can be seen, in sensor 1 (Figure 4a) that there are some outliers that form a straight line at 687 hPa, and in sensor 2 (Figure 4b) that there is greater dispersion of the data. In all sensors the RMSE is less than 0.16 hPa, except for sensors 1 and 2 which have both 0.31 hPa. In the evaluation of the linear regression, it can be seen that the slope is almost perfect in all cases without exception (a > 0.97). On the other hand, the intercept varies from 0.85 in sensor 5 (Figure 4e) to 19.7 in sensor 7 (Figure 4g). The statistical results of all the sensors are shown in Table 2.

#### 3.1.3. Relative Humidity

All GMX500 sensors appear to be fairly close in relative humidity inter-comparison (Figure 5). Sensor 1 (Figure 5a) presents outliers in the same way as in pressure (Figure 4a), which apparently decreases its R${}^{2}$ to 0.946. All other sensors (Figure 5b–i) have an R${}^{2}$ greater than 0.99. In Figure 5 it can be observed that all the sensors have a similar data dispersion, with the exception of sensor 5. This sensor has greater dispersion in our data with an RMSE equal to 1.64, only surpassed by sensor 1 due to its outliers. The RMSE of the other sensors (i.e., Figure 5b–d,f–i) are less than 1.2. The linear regression of all sensors appears to be perfect to the naked eye, the slope values range from 0.98 to 1.01 and the intercept values range from 1.72 to −0.6. The statistical results of all the sensors are shown in Table 3.

#### 3.1.4. Wind Velocity

The inter-comparison of the wind speed shows much more dispersed data than in the previous variables (Figure 6). Despite the dispersion in the data, all the sensors seem to behave the same way. Sensor 1 returns to showing outliers on a vertical straight line near 3 m s${}^{-1}$ (Figure 6a). For this variable, the R${}^{2}$ in all sensors is quite low compared to the other variables. Sensor 1 has the lowest R${}^{2}$ (0.724), but just slightly lower than sensor 5 (0.728). Sensor 9 has the highest R${}^{2}$ with 0.83. The RMSE of all instruments is similar. Sensor 1 and 5 have the highest RMSE with 0.567 m s${}^{-1}$ (Figure 6a,e). While the lowest RMSE is from sensor 9 with 0.446 m s${}^{-1}$ (Figure 6i). Linear correlation analysis shows a low slope in all cases. Sensor 1 has the lowest slope at 0.745, while Sensor 7 has the highest slope at 0.814. The intercept in all cases is similar, sensor 9 has the lowest intercept (0.278) and the highest is that of sensor 5 (0.341). The statistical results of all the sensors are shown in Table 4.

### 3.2. Evaluation of GMX500 with a Conventional Weather Station

The GMX500 sensors were evaluated using a conventional weather station as a reference. The conventional station is under the administration of the National Service of Meteorology and Hydrology of Peru (SENAMHI). The data from the conventional weather station is taken manually by a trained observer, following the guidelines of the World Meteorological Organization (WMO). Conventional data is taken three times a day at 7, 13, and 19 h local time (UTC-5). In the first observation, the maximum and minimum temperature of the previous day is recorded. The variables to be evaluated are temperature, pressure and relative humidity. Wind speed data from the conventional station were not available. For the evaluation of the GMX500 sensors, data from the h closest to those recorded in the conventional station were used. The maximum and minimum temperature data were found as the maximum values between 7 h of the day in question and 7 h of the following day, in such a way that they are equivalent to conventional data.

Figure 7 shows the comparison of sensor 01 with the conventional weather station. The maximum and minimum temperature are the variables that have the best agreement (Figure 7a,b). The R${}^{2}$ is 0.93 and 0.92, for the maximum and minimum temperature, respectively. The RMSE of the maximum and minimum temperatures are 0.46 and 0.60, respectively. The linear regression analysis on the maximum and minimum temperature shows a slope of 0.99 and 0.97, respectively. The intercept for the maximum temperature is −0.05 and 0.42 for the maximum temperature. The statistical results of the temperature of 7, 13 and 19 h are shown in Table A1. The R${}^{2}$ is 0.90, 0.88, and 0.85 for 7, 13, and 19 h, respectively, slightly lower than the maximum and minimum temperatures. The RMSE of the temperature in those hours is 0.51, 0.72, and 0.76. The slope in the linear regression is 0.94, 0.95, and 1.02, respectively, while the intercept is 0.4, 1.2, and −0.3, respectively. The comparison of the relative humidity of sensor 01 with the conventional station is shown in Figure 7c. The R${}^{2}$ of relative humidity is 0.90 and the RMSE is 4.34. In the linear regression, the slope is 0.98 and the intercept is −11, indicating a slight offset. The comparison of the pressure of sensor 01 with the conventional station is shown in Figure 7d. The R${}^{2}$ of pressure is 0.78 and the RMSE is 0.57. In the linear regression, the slope is 1.08 and the intercept is −51.7. Pressure also has an offset similar to that of relative humidity.

The statistical results of the evaluation of sensors 02 to 09 with the conventional weather station are attached in Appendix A. The temperature of all sensors has a similar behavior to that of sensor 01. The maximum temperature is the variable with the highest agreement in all the sensors. The minimum temperature shows a slightly larger RMSE (around 0.61 °C) than the maximum temperature (around 0.43 °C). Furthermore, the intercept is also slightly higher at the minimum temperature (around 0.5 °C) than at the maximum temperature (around 0.07 °C). The temperature at 7 h has less dispersion (around 0.51 °C) than the temperature at 13 h (around 0.68 °C) and than the temperature at 19 h (around 0.76 °C). In linear regression, the slope at 07 h (around 0.96) is very similar to that at 13 h (around 0.95) and slightly lower than at 19 h (around 1.02). While the intercept at 7 h is very close to 0 (around 0.1), at 13 h it is very close to 1 in all sensors, and at 19 h it is approximately −0.3. The pressure in all the sensors behaves similarly to sensor 01, however, there are notable differences in the time of measurement (Table A2). The R${}^{2}$ at 19 h (around 0.59) is lower than at 7 h (around 0.78) and at 13 h (around 0.77). The RMSE at 19 h is higher than at 7 h (around 0.58) and at 13 h (around 0.57). In the linear regression, the slope per at 7 h is greater than 1 per around 0.24, while at 13 and 19 h they do not exceed 1.1. The intercept shows greater diurnal dependence. At 7 h it is around −162 hPa, while at 13 h it drops to around −55.7 hPa, and at 19 h it is around −6.5 hPa. The relative humidity in all sensors also behaves similarly to sensor 01 (Table A3). The conventional sensor that measures relative humidity has a measurement range of up to 90%. At nightfall, when storms usually start in the area, the relative humidity can exceed 90% and even reach 100%, high relative humidity can last until dawn. The limited measurement range of the conventional sensor means that the evaluation at 7 and 19 h has worse results than at 13 h. The R${}^{2}$ for all sensors is on average 0.71, 0.76, and 0.86, for the relative humidity at 7, 13, and 19 h, respectively. The RMSE is on average 3.1, 0.6, and 7.5, at 07, 13, and 19 h, respectively. The slope is on average 0.62, 1.08, and 1.25, at 7, 13, and 19 h, respectively. The intercept is on average 31.2, −54.7, and −27.2 at 7, 13, and 19 h, respectively.

### 3.3. Analysis of WS100

The analysis of the WS100 sensors is divided into two parts. In the first part the estimation of precipitation is evaluated and in the second part the capacity of the instrument to measure the drop size distribution (DSD) is evaluated. For the comparison between the WS100 and Parsivel${}^{2}$ sensors, the data is separated by rain events. We use the [5] definition of rain event, as a period separated by at least 2 h of no-rain period, the rain/no-rain threshold was set as a minimum of 10 drops and a rain intensity of 0.1 mm h${}^{-1}$ using Parsivel${}^{2}$ as reference. We discard the events whose accumulated total rainfall is less than 1 mm. During the study period, a total of 23 rain events were found.

### 3.4. Rainfall Rate

The rain intensity is evaluated by 4 parameters: the total rain, the minutes of rain, the maximum intensity, and the distribution of the instantaneous rain rate. In Figure 8 shows the comparison of the events rain totals, the number of rainy minutes per event, and the event maximum intensity of WS100 versus Parsivel${}^{2}$. The statistical results are shown in Table 5. Linear regression has been restricted to calculating the slope because the intercept is assumed to be 0. In this case, the slope is equivalent to a correction factor.

The first thing that can be noticed from this comparison is that the accumulated rainfall (Figure 8a), despite the fact that there is sufficient similarity between the WS100 sensors, the total rainfall is higher than that registered by the Parsivel${}^{2}$. The percentage bias and the absolute percentage bias in all sensors are between 88% to 117% and 89% to 118%, respectively. Sensor 10 has the lowest bias and sensor 8 has the highest bias. However, the biases of sensor 10 are similar to those of sensor 9. All sensors except 8 and 10 have a correlation between 0.90 and 0.93 (Table 5). The sensor 8 has the lowest correlation (89%) and the highest correlation corresponds to the sensor 10 (0.97%). The slope of sensors 1 to 7 is between 2 and 2.2. Sensor 8 has the highest slope (2.3), while sensors 9 and 10 have the lowest slope (1.85).

The number of rainy minutes (Figure 8b) is less than that registered by the Parsivel${}^{2}$ in all sensors (average bias between −14% and −18%), except sensor 10 that registered slightly more minutes of rain (average bias of 4.4%). The absolute percent bias is about 2 to 3 percentage points higher than average percent bias in most sensors, except in sensor 10 (absolute bias 0.41 percentage points higher than average percent bias). The correlation is almost perfect in all sensors, which is 0.99 in sensors 1 to 9, while sensor 10 has a perfect correlation (correlation equal to 1). The slope is almost the same in sensors 1 to 9 (approximately 0.85), while in sensor 10 it is 1.28 (Table 5).

In the maximum intensity registered (Figure 8c), the average percentage bias in the sensors ranges from 139% to 172%. Sensor 9 has the lower percent bias and the sensor 2 has the highest percentage bias. The absolute percentage bias is similar to the percentage bias in all sensors without exception (between 1 and 7 percentage points higher). The linear correlation in all sensors ranges between 0.81 and 0.89, the sensor 9 has the lower correlation and the sensor 2, 8 and 10 have the higher correlation. The slope in the maximum intensity ranges from 2.0 to 2.8, the sensor 10 has the lower slope and the sensor 2 has the higher slope (Table 5).

The statistical distribution of the instantaneous rain rate is shown in Figure 9. The probability distribution (Figure 9a), shows that all the sensors have problems registering values lower than 0.1 mm h${}^{-1}$, except sensor 10 which is more similar to Parsivel${}^{2}$. The minimum intensity measured by Parsivel${}^{2}$ is 0.01 mm h${}^{-1}$. Sensors 1 to 9 present peaks in their distribution that stand out especially between 0.1 and 0.2 mm h${}^{-1}$, and another between 0.5 and 0.6 mm h${}^{-1}$. The pikes are caused by the low sensitivity of the sensors. Sensor 10 has a pair of peaks at lower intensities, one between 0.01 and 0.02 mm h${}^{-1}$ and the other between 0.06 and 0.07 mm h${}^{-1}$. All the sensors have more agreement to Parsivel${}^{2}$ between the intensities of 2 to 5 mm h${}^{-1}$; after that value the WS100 sensors register higher intensities. It can be seen in Figure 9 that the highest intensity values are much more frequent than they should be, which means that the WS100 sensors record more precipitation in total rainfall and have a higher maximum intensity.

### 3.5. Drop Size Distribution (DSD)

The DSD data output from the WS100 sensors consists of a table with the number of drops for each range of sizes. The WS100 sensor has 11 drop size classes ranging from 0.3 to 5.0 mm. The drop size class width is fixed in 0.5 mm, and the last class is reserved to drops greater than 5 mm. The sampling output step of 1 min is used in this work. The number format in the output data is integers and only the number of drops is recorded for each class. The main difficulty in comparing DSD data with WS100 sensors is the inability to represent DSD in standard units, such as number of drops per class width and per volume (i.e., mm${}^{-1}$ m${}^{-3}$). According to the manufacturer, the WS100 sensor’s operating principle is based on a Doppler radar. Unfortunately, they do not provide enough information on the theoretical bases and technical details necessary to handle the units of the DSD. Notice that the DSD retrieval using a vertically pointing Dopper radar is widely documented in the literature (e.g., [9,15,16]).

The Figure 10 show the comparison of the DSD retrieved by the Parsivel${}^{2}$ and WS100 sensors expressed in number of drops for 23 rain events. The WS100 sensors register very few drops for sizes smaller than 2 mm and more drops for drop sizes greater than 4 mm. The curves intersect in the drops of medium size, between 2 and 3 mm. In Parsivel${}^{2}$ the peaks of the drop concentration are close to 0.5 mm. The distribution in all cases resembles the Gamma DSD model [17]. The DSD recorded in by the WS100 sensors is atypical, because the curve is almost constant from 1 to 5 mm. All profiles show a slight decrease in the amount of drops in the range of 2 to 4 mm of drop size. The number of drops greater than 4 mm, registered by the WS100 sensors, is greater than what would be expected, so hat it can be considered as unrealistic.

## 4. Discussion

The results of the inter-comparison of the GMX500 sensors show that sensors 1, 2, and 10 present a considerably higher data dispersion than the others, in many cases the RMSE could be up to 5 times higher (see Table 1, Table 2, Table 3 and Table 4). The data dispersion was spatially noticeable in the comparison of the temperature (Figure 3), however, the pressure, relative humidity and wind speed showed outliers (Figure 4, Figure 5 and Figure 6) that suggested that something in the instrument configuration was wrong. Consequently, we decided to increase the study period by carrying out some tests in the configuration within the data loggers during the period of February–March 2021. Fortunately, it was possible to correct the errors caused by a bad configuration of the instruments, however a large amount of data had to be discarded for sensors 1, 2, and 9. Figure 11 shows the inter-comparison of sensor 01 after correcting the settings in the data logger. The authors find it important to show how small changes in the configuration can cause measurement errors that can be imperceptible when observing the data in real time. It is important to check that the scripts are identical. With the optimized configuration, all GMX500 sensors showed great similarity to each other. Sensors RMSE for temperature does not exceed 0.3 °C, for pressure the RMSE is less than 0.1 HPa, for relative humidity the RMSE is less than 1%, except for sensor 5, whose RMSE is 1.3%, and for wind velocity the RMSE is less than 0.6 m s${}^{-1}$. The wind velocity presents higher variability that other variables, due to it high temporal variability influenced by the turbulence.

In the evaluation of the GMX500 sensors with the conventional weather station, they show that the maximum and minimum temperature have better agreement with the conventional station. Statistical results are different depending on the time of day. It is likely that conventional sensors have a longer response time than automatic sensors or that the observer’s measurements, which can take several minutes, explain the differences between the instruments at specific times. Notice that the maximum and minimum temperatures do not need to be recorded at the same time by the instruments. Probably better results can be obtained by doing a cross-correlation analysis. For relative humidity, the limited range of the conventional sensor makes it difficult to compare between night and dawn, where humidity is often very high.

The event-evaluated rainfall estimate shows that all WS100 sensors overestimate rainfall. The average bias and absolute bias are around 100%, only sensors 8 and 9 have less bias (87% and 88%, respectively). The WS100 sensors record fewer rainy minutes than the Parsivel${}^{2}$, possibly due to lower sensitivity (Figure 8b and Figure 9). The sensor 10 has shown to have a sensitivity similar to that of Parsivel${}^{2}$ and in general terms it has a better estimation of precipitation. Based on how the rainfall intensities are distributed it can be seen that the WS100 sensors have problems in estimating low rainfall intensities. Almost all sensors, with the exception of sensor 10, do not register intensities lower than 0.1 mm h${}^{-1}$. For intensities lower than 2 mm h${}^{-1}$, peaks were found in the distribution, which would be attributed to the low sensitivity of the instruments. WS100 sensors perform best at intensities between 2 and 5 mm h${}^{-1}$ and have the best agreement with Parsivel${}^{2}$ measurements. At intensities greater than 5 mm h${}^{-1}$, the WS100 sensors overestimate the precipitation, which explains why the error in the maximum intensities is greater than in accumulated rainfall. The slope calculated in the precipitation analysis cannot be used as a correction factor due to the different behavior that the WS100 sensors present at different intensities. The DSD retrieved from the WS100 sensors is very different from that of the Parsive${}^{2}$, the similar concentrations in 1 mm and 5 mm suggest that there is an error in the estimation of the number of drops, in addition, the flat curve of drop contractions that are observed in all the events is unrealistic (Figure 10). With the information available in the data output, it is not possible to correct the DSD. The authors have requested more information on the technical details of the operation and theoretical bases of the WS100 sensors, which would allow correcting all the rainfall estimation parameters. Correction of the DSD of the WS100 sensors and further analysis of the DSD estimate remain for future work.

## 5. Conclusions

In the present work, two sensors that are part of a compact meteorological station have been tested. The GMX500 sensor, whose evaluated variables were: temperature, pressure, relative humidity and wind speed. And the WS100 sensor that is designed to estimate rain and DSD through Doppler radar. With data collected in the period from October 2020 to January 2021, 9 GMX500 sensors were compared. The average of the 9 instruments was used as a reference. In the first evaluation it was found that 2 sensors had a lot of data dispersion with an RMSE up to 5 times higher than the rest. It was found that the errors came from a bad configuration in the data loggers, which led to the extension of the analysis period with different tests between February and March 2021. With the new configuration, it was achieved that all the sensors have the same behavior and the RMSE of the different variables were not greater than 0.3° C for temperature, not greater than 0.1 hPa for pressure, and not greater than 1% for relative humidity (except sensor 5 with 1.3%), and not higher than 0.6 m s${}^{-1}$ for wind speed. The coefficient of determination R${}^{2}$ for all GMX500 sensors were: greater than 0.99 for temperature, greater than 0.99 for pressure and for relative humidity, and between 0.72 and 0.81 for wind speed. It has been observed that sensor 5 has slightly lower performance in relative humidity and wind speed, but the causes could not be determined. The GMX500 sensors were also evaluated with data from a conventional weather station. The best results were found in the maximum and minimum temperatures. The comparison was made independently for 7, 13, and 19 h. Different results were found depending on the time of day. The differences are possibly influenced by the response time of conventional sensors to environmental changes or by the minutes that the observer may take to record the data, which would explain why better results are obtained in the maximum and minimum temperatures. The WS100 sensors were evaluated using a Parsivel${}^{2}$ optical disdrometer as a reference. In the period from October 2020 to January 2021, 9 WS100 sensors were used and one sensor was added in the period of October 2020 to January 2021, as part of a collaboration with the CEPREANDES project. All WS100 sensors were found to overestimate precipitation, with a percent bias close to 100%. The absolute bias is slightly higher with up to 2 percentage points above the percentage bias. It was observed that the differences in the estimates with Parsivel${}^{2}$ increase with greater intensity of rain. The minimum intensity detected by the sensors is 0.1 mm h${}^{-1}$, except for sensor 10 which resembles that of Parsivel${}^{2}$ with 0.01 mm h${}^{-1}$. For intensities lower than 2 mm h${}^{-1}$, peaks were found in the distribution, which would be attributed to the low sensitivity of the WS100 instruments. At intensities between 2 and 5 mm h${}^{-1}$ they have the best agreement with Parsivel${}^{2}$. At intensities greater than 5 mm h${}^{-1}$, the WS100 sensors overestimate the precipitation, which explains why the error in the maximum intensities is greater than in accumulated rainfall. At intensities lower than 2 mm h${}^{-1}$, subestimation of precipitation could be expected due to the low sensitivity of the instruments and an overestimation at intensities higher than 5 mm h${}^{-1}$. Due to the behavior of different intensities it is not possible to apply a correction factor. The DSD retrieved by the WS100 sensors have an unrealistic behavior with higher concentrations in diameters of 1 mm and 5 mm, in addition to presenting a flattened curve. The DSD retrieved by the WS100 sensors have an unrealistic behavior with higher concentrations in diameters of 1 mm and 5 mm, in addition to presenting a flattened curve. This suggests that the WS100 sensors have errors in the estimation of the number of drops or in the data output. It is hoped that with more information provided by the manufacturers it will be possible to correct the DSD and the other rainfall parameters, which remains as future work.

## Figures and Tables

**Figure 1 sensors-22-03219-f001:**
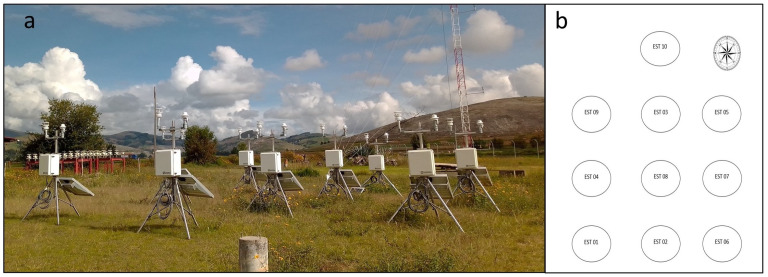
Distribution of the compact automatic stations in the site of study. (**a**) Picture of the installed compact automatic station in the Huancayo Observatory. (**b**) Distribution of the 10 stations, which have 3.5 m of the horizontal and vertical separation.

**Figure 2 sensors-22-03219-f002:**
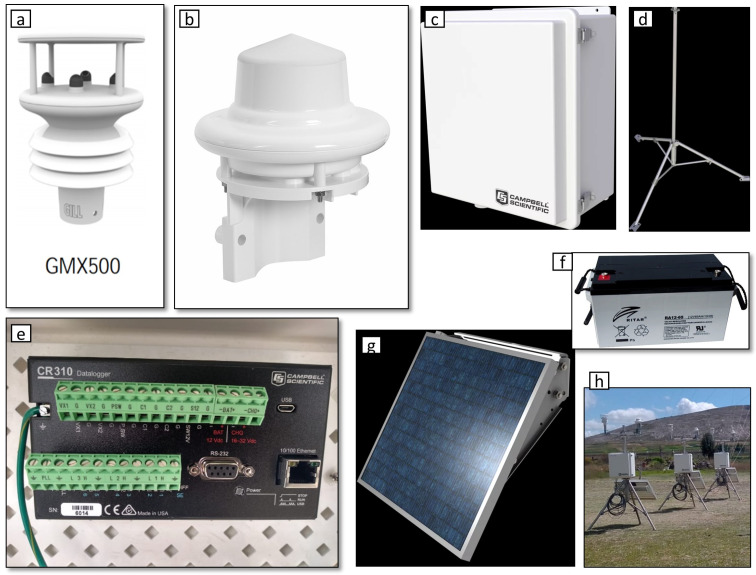
Components of the automatic compact stations. (**a**) Weather Sensor GMX500 MAXIMET GILL series. (**b**) WS100 Lufft Precipitation Sensor. (**c**) Campbell Scientific Box. (**d**) Stainless steel tripod. (**e**) Datalogger CR310—CAMPBELL SCIENTIFIC. (**f**) Battery Ritar 12 v 65 Ah AGM. (**g**) SP30 solar panel. (**h**) Final assembled stations.

**Figure 3 sensors-22-03219-f003:**
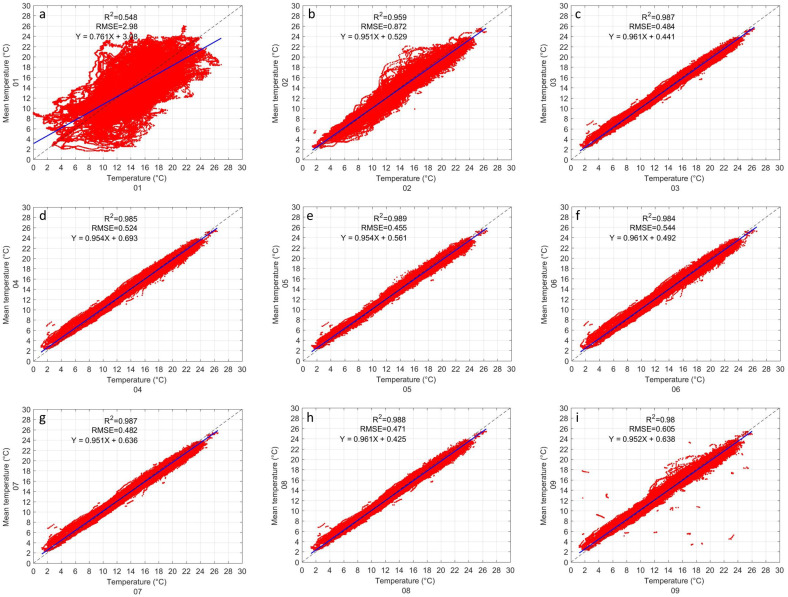
Inter-comparison of temperature between GMX500 sensors. Each panel (**a**–**i**) corresponds to a sensor under evaluation. The coefficient of determination (R${}^{2}$), the root mean square error (RMSE) and the lineal regression (in blue line) are shown in each panel.

**Figure 4 sensors-22-03219-f004:**
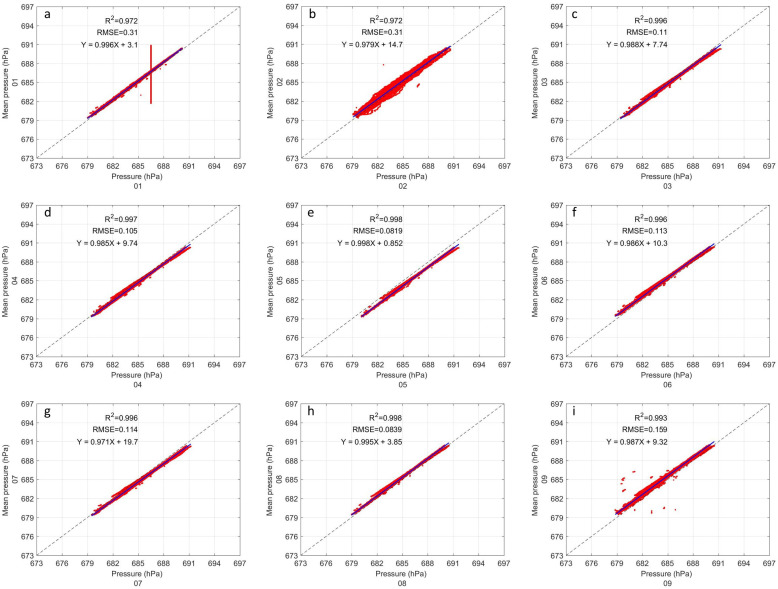
Same as Figure 3 but for pressure.

**Figure 5 sensors-22-03219-f005:**
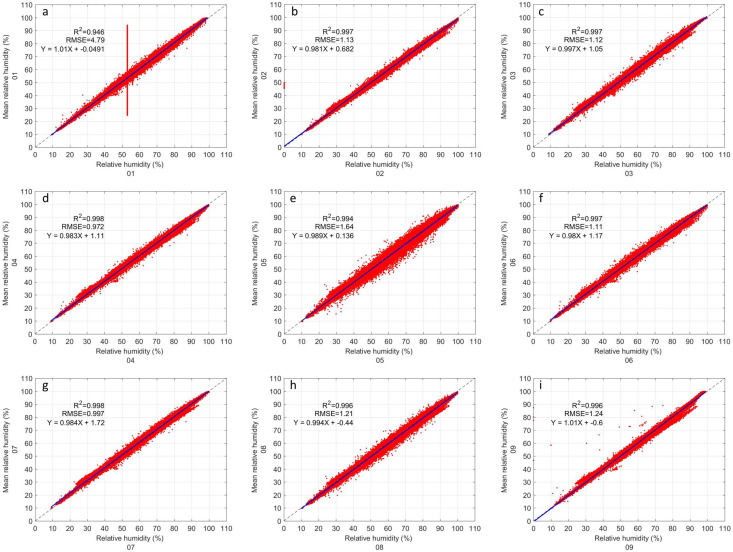
Same as Figure 3 but for relative humidity.

**Figure 6 sensors-22-03219-f006:**
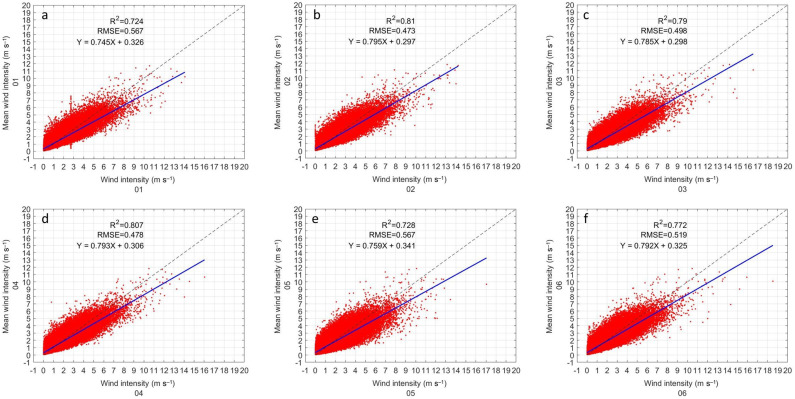
Same as Figure 3 but for wind velocity.

**Figure 7 sensors-22-03219-f007:**
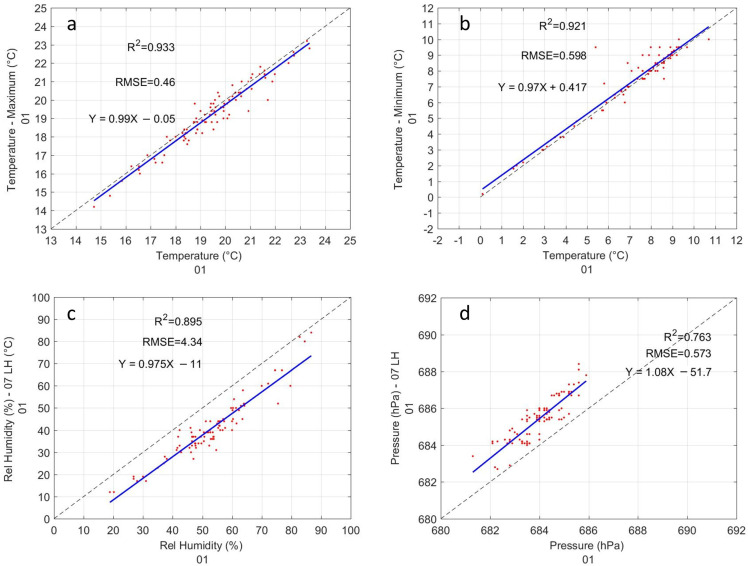
Evaluation of GMX500 sensor 01 with a conventional weather station: (**a**) Minimum temperature. (**b**) Maximum temperature. (**c**) Pressure. (**d**) Relative Humidity. The linear regression is shown in blue line.

**Figure 8 sensors-22-03219-f008:**
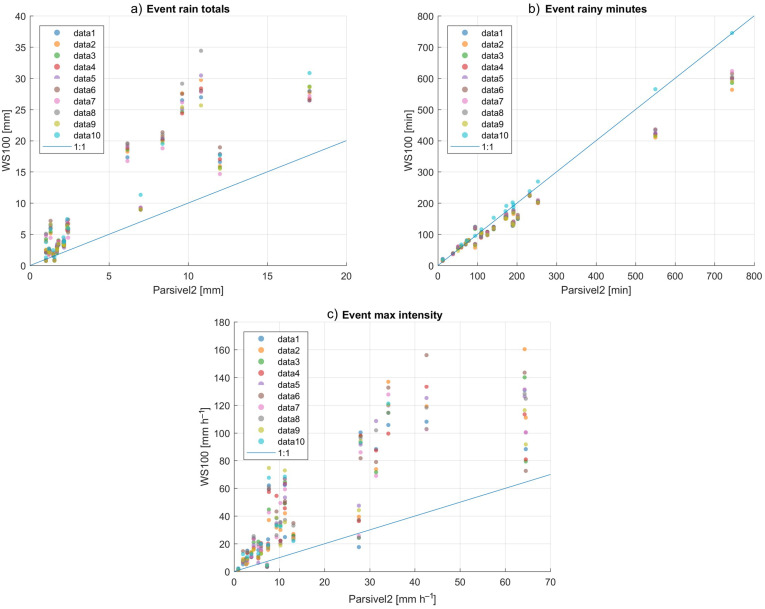
Comparison of ten WS100 sensors with Parsivel${}^{2}$ in rainfall events: (**a**) Accumulated event rain totals greater than 1 mm. (**b**) Duration of each event in minutes. (**c**) Maximum intensity registered by each event. Parsivel${}^{2}$ is used as reference and 23 events where found. The blue dashed line indicates the 1:1 relation.

**Figure 9 sensors-22-03219-f009:**
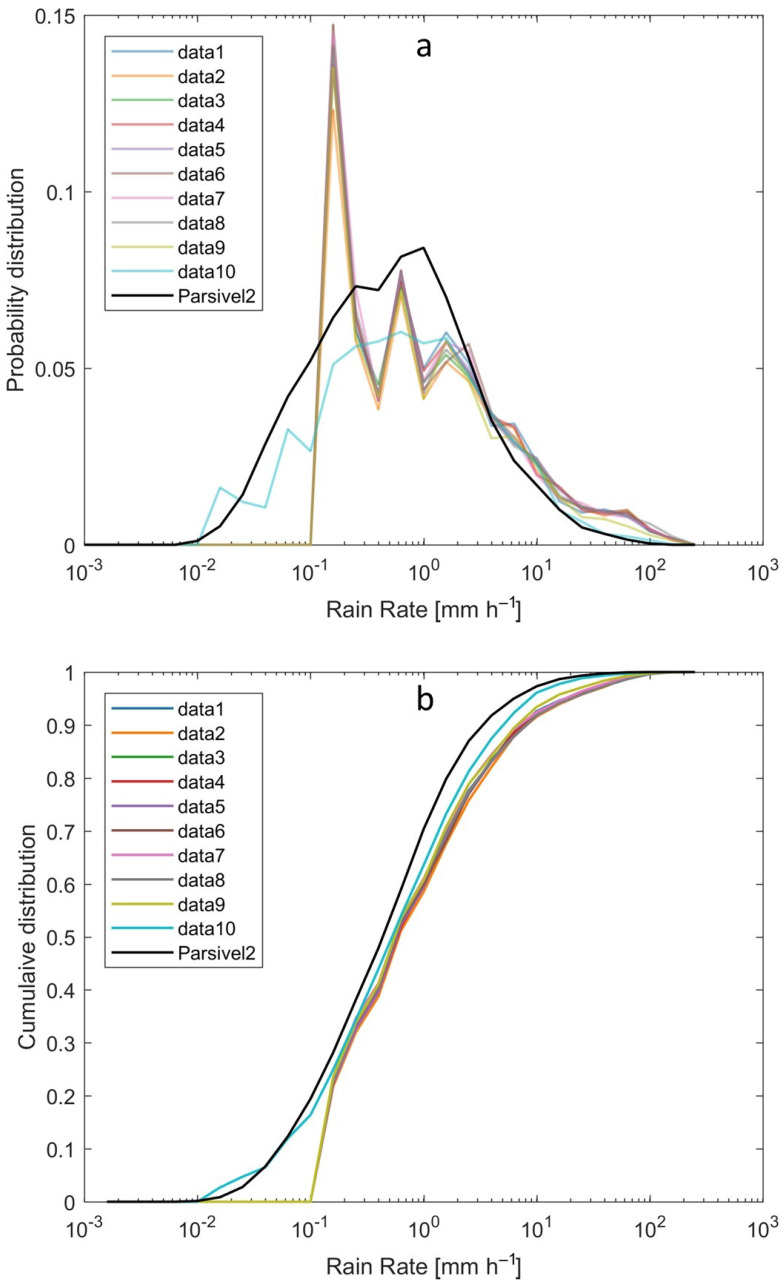
Statistical distribution of rain rate at 1 min sampling output. (**a**) Probability distribution of rain rate. (**b**) Cumulative distribution of rain rate. The Parsivel${}^{2}$ is shown in black line.

**Figure 10 sensors-22-03219-f010:**
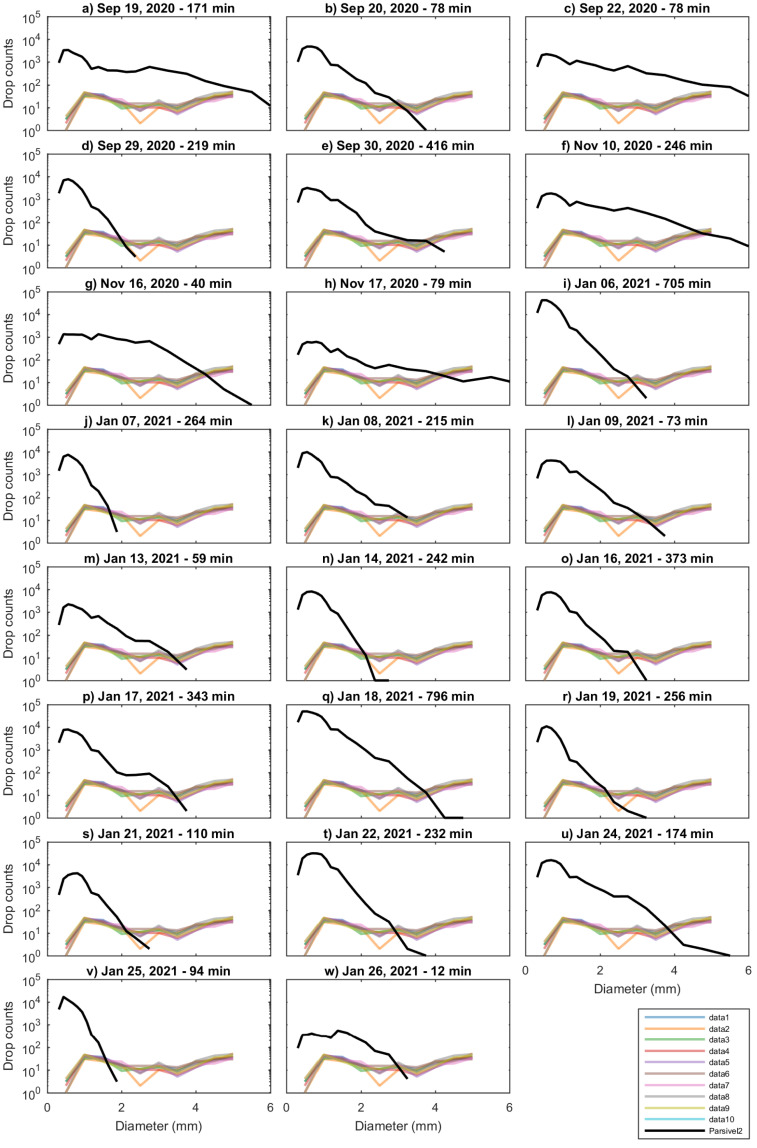
Drop size distribution measured by Parsivel${}^{2}$ (black line) and WS100 sensors (color lines) for 23 rainfall events. From (**a**–**w**), the DSD for each event is shown.

**Figure 11 sensors-22-03219-f011:**
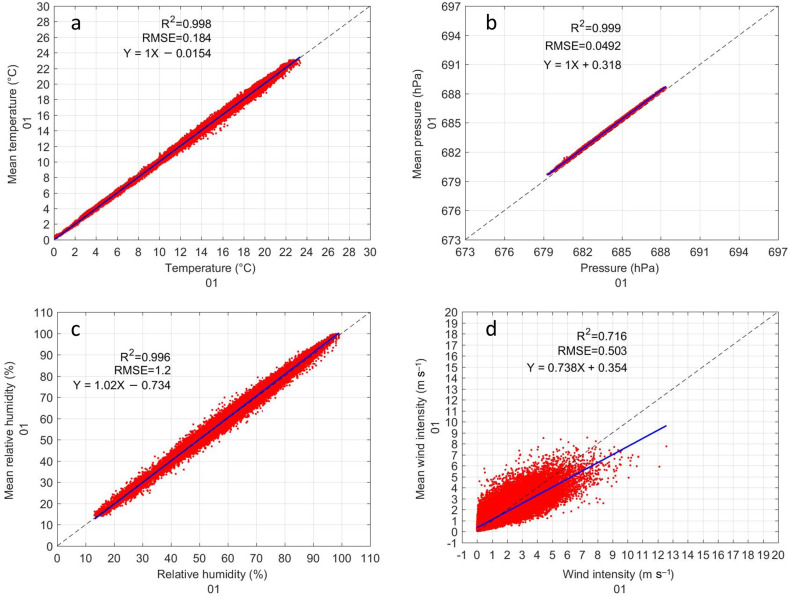
Inter-comparison of GMX500 sensor 01 after correcting the settings in the data logger: (**a**) Inter-comparison of temperature. (**b**) Inter-comparison of pressure. (**c**) Inter-comparison of relative humidity. (**d**) Inter-comparison of wind velocity.

**Table 1 sensors-22-03219-t001:** Statistical results of the inter-comparison of the temperature of the GMX500 sensors.

Temperature				
**Sensor**	**R** ${}^{2}$	**RMSE**	**Slope (a)**	**Intercept (b)**
1	0.548	2.98	0.761	3.080
2	0.959	0.872	0.951	0.529
3	0.987	0.484	0.961	0.441
4	0.985	0.524	0.954	0.693
5	0.989	0.455	0.954	0.561
6	0.984	0.544	0.961	0.492
7	0.987	0.482	0.951	0.636
8	0.988	0.471	0.961	0.425
9	0.980	0.605	0.952	0.638

**Table 2 sensors-22-03219-t002:** Statistical results of the inter-comparison of the Pressure of the GMX500 sensors.

Pressure				
**Sensor**	**R${}^{2}$**	**RMSE**	**Slope (a)**	**Intercept (b)**
1	0.972	0.310	0.996	3.10
2	0.972	0.310	0.979	14.70
3	0.996	0.110	0.988	7.74
4	0.997	0.105	0.985	9.74
5	0.998	0.0819	0.998	0.85
6	0.996	0.113	0.986	10.30
7	0.996	0.114	0.971	19.70
8	0.998	0.0839	0.995	3.85
9	0.993	0.159	0.987	9.32

**Table 3 sensors-22-03219-t003:** Statistical results of the inter-comparison of the Relative Humidity of the GMX500 sensors.

Relative Humidity				
**Sensor**	**R${}^{2}$**	**RMSE**	**Slope (a)**	**Intercept (b)**
1	0.946	4.790	1.010	−0.049
2	0.997	1.130	0.981	0.682
3	0.997	1.120	0.997	1.050
4	0.998	0.972	0.983	1.110
5	0.994	1.640	0.989	0.136
6	0.997	1.110	0.980	1.170
7	0.998	0.997	0.984	1.720
8	0.996	1.210	0.994	−0.44
9	0.996	1.240	1.010	−0.600

**Table 4 sensors-22-03219-t004:** Statistical results of the inter-comparison of the Wind Velocity of the GMX500 sensors.

Wind Velocity				
**Sensor**	**R${}^{2}$**	**RMSE**	**Slope (a)**	**Intercept (b)**
1	0.724	0.567	0.745	0.326
2	0.810	0.473	0.795	0.297
3	0.790	0.498	0.785	0.298
4	0.807	0.478	0.793	0.306
5	0.728	0.567	0.759	0.341
6	0.772	0.519	0.792	0.325
7	0.806	0.479	0.814	0.280
8	0.772	0.518	0.790	0.321
9	0.831	0.446	0.788	0.278

**Table 5 sensors-22-03219-t005:** Statistical results of the comparison of the WS100 sensors and Parsivel${}^{2}$.

Event Rain Totals										
WS100 N°	1	2	3	4	5	6	7	8	9	10
Bias avg. (mm)	4.36	4.69	4.39	4.39	4.68	4.77	4.20	5.13	3.84	3.90
Bias avg. (%)	99.12	106.72	99.85	99.75	106.36	108.45	95.43	116.66	87.40	88.61
Bias abs. (%)	100.83	108.47	101.86	101.57	108.39	110.41	97.22	118.73	89.53	89.24
Correlation	0.92	0.90	0.92	0.91	0.90	0.91	0.91	0.89	0.93	0.97
Slope	2.102	2.237	2.114	2.141	2.186	2.235	2.067	2.313	1.855	1.852
**Event Rainy Minutes**										
WS100 N°	1	2	3	4	5	6	7	8	9	10
Bias avg. (min)	−25.52	−28.68	−26.83	−25.35	−26.00	−22.04	−22.04	−24.35	−31.43	7.53
Bias avg. (%)	−14.86	−16.70	−15.62	−14.76	−15.14	−12.84	−12.84	−14.18	−18.30	4.39
Bias abs. (%)	17.09	18.77	17.75	17.19	17.72	15.22	15.62	17.06	19.08	4.70
Correlation	0.99	0.99	0.99	0.99	0.99	0.99	0.99	0.99	0.99	1.00
Slope	0.851	0.825	0.844	0.852	0.849	0.872	0.872	0.858	0.876	1.279
**Event Max Intensity**										
WS100 N°	1	2	3	4	5	6	7	8	9	10
Bias avg. (mm/h)	26.05	30.92	26.04	27.23	29.98	29.97	27.10	30.38	24.08	25.50
Bias avg. (%)	150.55	178.68	150.50	157.38	173.24	173.21	156.61	175.58	139.18	147.35
Bias abs. (%)	157.33	180.30	154.16	159.24	175.24	174.91	158.56	179.09	140.66	149.49
Correlation	0.84	0.89	0.85	0.82	0.87	0.82	0.86	0.89	0.81	0.89
Slope	2.506	2.825	2.505	2.574	2.732	2.732	2.566	2.756	2.283	2.049

## Data Availability

The study campaign data used in this work can be found at doi: http://dx.doi.org/10.13140/RG.2.2.14043.64806 (accessed on 4 February 2022).

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
