# Peer review of "Field Campaign Evaluation of Sensors Lufft GMX500 and MaxiMet WS100 in Peruvian Central Andes"

_sensors, 2022, doi:10.3390/s22093219_

Round 1

Reviewer 1 Report

Comments to paper “Field campaign evaluation of sensors GMX500 and WS100 in Peruvian central Andesby JM Valdivia et al.

Title: In addition to models, Specify the producers Lufft WS100 and MaxiMet GMX500. In the abstract and in the text, use the models only.

Abstract: There are evidences of inappropriate use of written English. There are syntax errors which need to be fixed by a mother tongue.

The term intercomparison is normally used when comparing results from different techniques measuring the same variables. In this case, AA are comparing the results of an instrument with another instrument assumed as reference or with the average of many. This is an evaluation, as it is correctly reported in the title and not an intercomparison.

Acronyms (DSD) should be avoided since it is defined later in the text. here use the full meaning: Drop Size Distribution)

Text

Line

Comments

19

Give references of the producers such as address, web site, etc.

27

Temperature is missing

31

Give the references of OTT

46

Give reference to the statement about capability of providing “similar information”

74

This paragraph is incomplete

77

Give reference to the norms applied in the experiment (ISO?)

80

This subchapter includes many details which are not essential to the paper. Please be concise.

126

Laser band is inappropriate (laser beam?)

136

The GMX500 assume as reference the average of data provided by the instrument. This is not a correct approach since the comparison of a data with an average in which the data itself is contained does not provide information about quality in terms of accuracy. Try to compare the average excluding the sensor under evaluation against the data by this sensor

212

A correction factor requires that the physical reasons for the a correction are known. Otherwise, the data is statistically forced without any physical meaning. In addition, if the correction factor changes with time, the initial uncertainty will remain unchanged. This is the reason why the data should be stratified on time. It would be interesting to compare the field data with the producer specifications

248

This statement is not clear

Author Response

Dear reviewer,
we appreciate your important comments for the development of this article. Please review the attached document.

Reviewer 2 Report

This reviewer thanks the authors for a clear, succinct, focused, and well-organized paper reviewing the performance of weather parameter instrumentation sets. The paper documents a particular instrument set and would be useful to researchers looking to utilize that configuration – a small niche given the breadth of technology in the market. Since the aim of the paper is instrument performance, this reviewer suggests to the authors that they need to document some of the further rigour they would have thought about in terms of their verification and validation process.

Detail comments:

  • Line 12 – ‘I’ should be ‘it’.
  • Line 58 – The use of the Parsivel as a reference instrument needs discussion. Has it been laboratory benchmarked? Is it known to be a good reference instrument by others? Indeed, looking at Figure 9, the Parsivel instrument seems unsuitable to be a reference instrument (by ‘orders of magnitude’ according to the ‘Drop Count’ axis at smaller drop diameters). While the measurement regimes are further explained in the discussion section – the authors need to discuss or explain why they ‘stuck’ with it (logistics perhaps) – the authors could then make a recommendation from the noted limitation.
  • Line 74 – area specification offered – but not provided.
  • Line 77 – ‘meticulously carried out’ – by what standard? Or guide? Or leave the statement out as superfluous?
  • The authors should justify the use of instrument ensemble mean as a general instrument data validation method with reference to the literature. It seems to this reviewer that the comparison of ensemble mean is a ‘relative’ technique, not an ‘absolute’ technique. The use of some industry accredited laboratory calibrated reference instruments would be ‘absolute’. This may be suitable for the author’s needs – but it needs justification for the paper to be useful to other researchers. This item is a key change this reviewer seeks to the paper.
  • Line 185 – this reviewer would like to see the authors posit why the wind velocity data disparity exists as noted.
  • Some of the figures need to be enlarged so that legends and axis nomenclature can be read – figures 3, 4, 5. 6, 7, 9 and 10.
  • Line 284 – since the paper is about a certain instrument set capability it would be helpful if they explained the ‘bad configuration’ noted here – what lesson is learnt?

Author Response

(The authors gave the same response as above.)

Round 2

Reviewer 1 Report

Please improve some minor english style

Author Response

Dear reviewer,

we have improved the writing of English with an expert.

We appreciate your helpful comments.

Reviewer 2 Report

This reviewer thanks the authors for their comments and corrections - and also agrees with the discussion.

Author Response

(The authors gave the same response as above.)
